# OpenReview forum: "MaskTwins: Dual-form Complementary Masking for Domain-Adaptive Image Segmentation"
_ICLR.cc/2025/Conference — Submitted to ICLR 2025_

### Official Review · Reviewer_9P4o · 2024-11-02

**Soundness:** 2
**Presentation:** 3
**Contribution:** 2
**Rating:** 5
**Confidence:** 4

**Summary:**

This paper addresses the image segmentation problem in the context of unsupervised domain adaptation (UDA). Instead of using the random masks, authors design a dual-form complementary masked images to enhance the generalization of the overall framework. Authors argue that robust, domain-invariant features by enforcing consistency learning upon masked images can be learned. Extensive experiments on six experiments demonstrate the effectiveness of the proposed method.

**Strengths:**

-- The presentation of this paper is clear.

-- The idea of this paper is easy to understand.

-- The results look good on multiple datasets.

**Weaknesses:**

-- It is difficult to justify the contribution of this paper. The overall framework is based on many well-structured techniques, such as AdaIN, EMA-based pseudo label, consistency learning loss, and so on. There lacks an effective ablation study to clarify the gain that is actually taken by the envisioned dual-form complementary masked method. In the ablation study section, only the patch size and the mask type and mask ratio are reported.

-- The contribution is a bit incremental and over-claimed. At a high level, the envisioned dual-form complementary masked method can be regarded as the constrained entropy minimization like MEMO (Test Time Robustness via Adaptation and Augmentation) . Meanwhile, it seems that the proposed dual-form complementary masked method can also be applied to general classification problem, where we can mask the image in a same manner.  This can verify the authors' first contribution, "This perspective bridges the gap between masked image modeling and signal processing theory, potentially opening new avenues for future research". Applying the proposed method to more general tasks can further show the effectiveness of the proposed method.

-- In the introduction section, the authors clarify that "This insight is grounded in the principles of compressed sensing" and "We provide a theoretical foundation for masked reconstruction by reframing it as a sparse signal reconstruction issue". However, in the method section, there is no any presentation about these descriptions, which is very vague. Authors may reorganize the manuscript carefully.

**Questions:**

Please the weakness.

---

> ### Author Response · Authors · 2024-11-22
>
> Thank you for your valuable time and comments. The main concerns are addressed below.
>
> > **W1:** Ablation study of the used components
>
> Regarding your concerns, we fully ablate the used components on MitoEM-H → MitoEM-R in the following tables.
>
> ---
>
> Table R1. Ablation study of each loss component on MitoEM-H → MitoEM-R. The mean and standard deviation are computed over 3 random seeds. $L_{sup}$ = supervised loss, $L_{cl}$ = consistency loss, $L_{cm}$ = complementary masked loss, $L_{rm}$ = randomly masked loss, with a mask ratio of 0.5.
>
> |       | mAP       | F1        | MCC       | IoU       |
> |-------|-----------|-----------|-----------|-----------|
> | $L_{sup}$  | 96.38(±0.18) | 88.95(±0.07) | 88.60(±0.07) | 80.11(±0.11) |
> | $L_{sup}+L_{cl}$ | 96.60(±0.35) | 89.27(±0.10) | 88.94(±0.12) | 80.64(±0.17) |
> | $L_{sup}+L_{cl}+L_{rm}$ | 96.84(±0.10) | 89.80(±0.03) | 89.45(±0.04) | 81.49(±0.04) |
> | $L_{sup}+L_{cl}+L_{cm}$ | 96.87(±0.06) | 90.03(±0.06) | 89.66(±0.04) | 81.88(±0.09) |
> ---
>
> As shown in Table R1, adding the consistency loss $L_{cl}$ improves the performance over the supervised loss. Further, we separately incorporate the randomly masked loss $L_{rm}$ and the complementary masked loss $L_{cm}$. The results indicate that both losses contribute to performance improvements, with our proposed complementary masking strategy being more effective than the random masking strategy. The results with $L_{rm}$ and $L_{cm}$ have been visually shown in Figure 4 in the main paper.
>
> ---
>
> Table R2. Ablation study of EMA and AdaIN on MitoEM-H → MitoEM-R. The mean and standard deviation are computed over 3 random seeds.
>
> |       | mAP       | F1        | MCC       | IoU       |
> |-------|-----------|-----------|-----------|-----------|
> | Ours w/o  EMA & AdaIN | 96.74(±0.08) | 89.61(±0.05) | 89.23(±0.04) | 81.18(±0.08) |
> | Ours w/o EMA  | 96.85(±0.15) | 89.74(±0.02) | 89.38(±0.04) | 81.40(±0.04) |
> | Ours w/o AdaIN | 96.89(±0.14) | 89.88(±0.05) | 89.52(±0.03) | 81.63(±0.08) |
> | Ours | 96.87(±0.06) | 90.03(±0.06) | 89.66(±0.04) | 81.88(±0.09) |
> ---
>
> In our work, we adopt the EMA teacher to keep consistent with previous methods, such as CAMix [1], MIC [2], DAFormer [3], etc. The results show that both EMA and AdaIN contribute to performance improvements, with EMA having a more significant impact. To address any potential misunderstandings caused by the use of these components, we will include these experiments and necessary descriptions in the revised paper.

---

> > ### Author Response · Authors · 2024-11-22
> >
> > > **W2:** Connection with MEMO and application to classification tasks
> >
> >
> > 1. **Relation to MEMO:**
> > The constrained entropy minimization perspective of MEMO [4] indeed aligns with the broader concept of consistency regularization. However, our work differs in the following aspects:
> >    - MEMO applies augmentations, while **we focus on complementary masking as a novel form of augmentation grounded in theoretical principles of sparse signal reconstruction.**
> >    - Our dual-form complementary masked method ensures that the masked views are **not only complementary but also interpretable and optimized for learning domain-invariant features**, which goes beyond the scope of MEMO's augmentation-based approach.
> >    - We will cite MEMO [4] and briefly discuss this connection in the revised paper to strengthen the theoretical positioning of our work.
> >
> > 2. **Application to classification tasks:**
> > While our main focus is pixel-wise segmentation tasks, we agree that extending our method to classification tasks can further validate its effectiveness. To address this, we conducted additional experiments on the VisDA-2017 classification benchmark, as shown below.
> >
> > ---
> >
> > Table R3. Image classification accuracy in % on VisDA-2017 for UDA with ViT-B/16. The results are adopted from [2]. The mean and standard deviation are computed over 3 random seeds.
> >
> > |Method| Plane | Bcycl | Bus  | Car  | Horse | Knife | Mcytle | Person | Plant | Sktb | Train | Truck | Mean  |
> > |----|---|-------|------|------|-------|-------|--------|--------|-------|------|-------|-------|-------|
> > | TVT  | 92.9  | 85.6 | 77.5 | 60.5  | 93.6  | 98.2   | 89.3   | 76.4  | 93.6 | 92.0  | 91.7  | 55.7  | 83.9  |
> > | CDTrans | 97.1 | 90.5 | 82.4 | 77.5  | 96.6  | 96.1   | 93.6   | 88.6  | 97.9 | 86.9  | 90.3  | 62.8  | 88.4  |
> > | SDAT | 98.4 | 90.9 | 85.4 | 82.1  | 98.5  | 97.6   | 96.3   | 86.1  | 96.2 | 96.7  | 92.9  | 56.8  | 89.8  |
> > | SDAT w/ MAE | 97.1 | 88.4 | 80.9 | 75.3  | 95.4  | 97.9   | 94.3   | 85.5  | 95.8 | 91.0  | 93.0  | 65.4  | 88.4  |
> > | MIC [2]  | 99.0 | 93.3 | 86.5 | 87.6  | 98.9  | 99.0   | 97.2   | 89.8  | 98.9 | 98.9  | 96.5  | 68.0  | 92.8  |
> > | Ours | **99.1**(±0.0) | **95.0**(±1.1) | **86.6**(±1.1) | **89.0**(±0.7) | 98.8(±0.1) | **99.3**(±0.1) | 96.8(±0.4) | 88.3(±0.3) | 98.8(±0.1) | **99.1**(±0.1) | **97.2**(±1.7) | **69.7**(±0.9) | **93.1**(±0.1) |
> > ---
> >
> > Table R4. Image classification accuracy in % on VisDA-2017 for UDA with ResNet-101. The results are adopted from [2]. The mean and standard deviation are computed over 3 random seeds.
> >
> > |Method| Plane | Bcycl | Bus  | Car  | Horse | Knife | Mcytle | Person | Plant | Sktb | Train | Truck | Mean  |
> > |----|---|-------|------|------|-------|-------|--------|--------|-------|------|-------|-------|-------|
> > | CDAN  | 85.2   | 66.9   | 83.0  | 50.8  | 84.2  | 74.9  | 88.1   | 74.5   | 83.4   | 76.0   | 81.9   | 38.0   | 73.9  |
> > | MCC    | 88.1   | 80.3   | 80.5  | 71.5  | 90.1  | 93.2  | 85.0   | 71.6   | 89.4   | 73.8   | 85.0   | 36.9   | 78.8  |
> > | SDAT   | 95.8   | 85.5   | 76.9  | 69.0  | 93.5  | 97.4  | 88.5   | 78.2   | 93.1   | 91.6   | 86.3   | 55.3   | 84.3  |
> > | MIC [2] | 96.7   | 88.5   | 84.2  | 74.3  | 96.0  | 96.3  | 90.2   | 81.2   | 94.3   | 95.4   | 88.9   | 56.6   | 86.9  |
> > | Ours | **96.9**(±0.4) | **88.8**(±0.8) | 81.8(±0.3) | **77.1**(±2.2) | **96.4**(±0.1) | 97.2(±0.2) | **90.3**(±1.3) | **83.8**(±0.5) | 93.3(±0.6) | 94.8(±1.0) | **90.2**(±0.5) | **57.4**(±4.1) | **87.3**(±0.1) |
> > ---
> >
> > The results show that our method improves the UDA performance by +0.3 and +0.4 percent points when used with a ViT and ResNet network, respectively. The improvement is consistent over almost all classes. Regarding the additional experiments on classification tasks, we appreciate your interest and we will include these results in the supplementary materials of the revised paper.

---

> > > ### Author Response · Authors · 2024-11-22
> > >
> > > > **W3:** Vague description of claim in the introduction.
> > >
> > > First, we have connected with compressed sensing in the method section.  Although we did not explicitly define the sparse signal modeling in the original manuscript, **Assumption 1 ("X = S + E + N")** represents a typical form of sparse signal modeling, which is widely recognized in compressed sensing theory. Specifically, the input image `X` is modeled as a combination of a sparse signal component (`S`), environmental factors (`E`), and additive Gaussian noise (`N`). This reflects "We provide a theoretical foundation for masked reconstruction by reframing it as a sparse signal reconstruction issue." Complementary masks (`D` and `1-D`) inherently act as sampling operators that ensure sufficient coverage of the signal space, aligning with the **compressed sensing principle** of reconstructing sparse signals from incomplete observations.
> > >
> > > In the revised manuscript, we will explicitly elaborate on this sparse signal modeling in the main text to make the connection with compressed sensing more explicit.
> > >
> > > Additionally, the detailed descriptions of theoretical proofs and analyses are presented in the supplementary material. For example, the proof of Theorem 5 in Appendix E has applied compressed sensing recovery guarantees in step 3, and the Restricted Isometry Property (RIP) is an important property in compressed sensing.
> > >
> > >
> > >
> > > **Reference**
> > >
> > > [1] Zhou, Qianyu, et al. "Context-aware mixup for domain adaptive semantic segmentation." IEEE Transactions on Circuits and Systems for Video Technology 33.2 (2022): 804-817.
> > >
> > > [2] Hoyer, Lukas, et al. "MIC: Masked image consistency for context-enhanced domain adaptation." Proceedings of the IEEE/CVF conference on computer vision and pattern recognition. 2023.
> > >
> > > [3] Hoyer, Lukas, Dengxin Dai, and Luc Van Gool. "Daformer: Improving network architectures and training strategies for domain-adaptive semantic segmentation." Proceedings of the IEEE/CVF conference on computer vision and pattern recognition. 2022.
> > >
> > > [4] Zhang, Marvin, Sergey Levine, and Chelsea Finn. "Memo: Test time robustness via adaptation and augmentation." Advances in neural information processing systems 35 (2022): 38629-38642.

---

> ### Comment · Reviewer_9P4o · 2024-11-22
> **Thank you and update**
>
> Dear Authors,
>
> Thanks for your efforts to take into account my comments.
>
> Unfortunately, it seems that the proposed method exhibits very marginal improvements based on Tables R1 and R2. Meanwhile, the overall framework is derived from many well-constructed modules, showing incremental contributions. Therefore, I maintain my score 5 and raise the confidence to 4.
>
> Thanks for your efforts again.

---

> > ### Author Response · Authors · 2024-11-24
> >
> > Thank you for your prompt feedback. But we respectfully cannot agree with the justifications here, and would like to further clarify as below:
> >
> > >“the proposed method exhibits very marginal improvements based on Tables R1 and R2”
> >
> > Table R1 and R2 are **intermediate** ablation results on our own method, not a direct comparison with **existing state-of-the-art (SoTA)** methods. As shown in Table 2 of the main paper, under the same setting with the ablation, our method notably outperforms the SoTA method (e.g., 81.9 VS 80.6 in IoU). The improvements under the other three settings are even more significant (e.g., 75.0 VS 71.8, 78.6 VS 75.4, and 78.4 VS 76.3). In our opinion, these results compared with the existing SoTA should be more informative to be used as evidence for performance justification, instead of the intermediate ablation results that may not be reported in literature before.
> >
> > >“the overall framework is derived from many well-constructed modules”
> >
> > If we compare Table R2 and Table 2 in the main paper, without AdaIN and EMA (the well-constructed modules), our method still achieves an IoU of 81.18, which already outperforms the **existing SoTA** (with an IoU of 80.6) and demonstrates the effectiveness of our key design (complementary masking). Moreover, in Table R1, adding the complementary masked loss to the **existing consistency loss** yields a notable improvement (from 80.64 to 81.88 in IoU). Therefore, while we use some well-constructed modules, the main performance improvement comes from the key contribution of complementary masking.
> >
> > We sincerely hope the above arguments can be taken into consideration of your final justification.

---

> > > ### Author Response · Authors · 2024-11-30
> > >
> > > Dear Reviewer 9P4o,
> > >
> > > As the discussion period is ending soon, we would like to send a kind reminder about our latest responses and the revised manuscript. We have addressed each of your concerns in detail and incorporated your suggestions into our revisions. If there are still remaining concerns, we will do our best to provide clarifications as soon as possible. Otherwise, we look forward to your positive feedback.
> > >
> > > Once again, we appreciate your time and consideration.

---

> > > > ### Comment · Reviewer_9P4o · 2024-12-01
> > > > **Thank you for your response**
> > > >
> > > > Dear Authors,
> > > >
> > > > Thanks for your responses. While I understand that Table R1 and R2 are intermediate ablation results on your method and are not a direct comparison with existing state-of-the-art (SoTA) methods, it seems that the performance improvement of the complementary masked loss is very marginal compared with that of random mask. For example, in Table R1, the IoU score of the the proposed complementary mask  is 96.87, slightly outperforming that of the random mask. i.e., 96.84. To me, the intermediate ablation study is much important to evaluate the contribution of the proposed method. Since the proposed complementary masked loss exhibits marginal gains by ablation study, I think it may be necessary to highlight other advantages, rather than achieving so-called SOTA performance.
> > > >
> > > > Therefore, I still remain the current score. If authors have additional justifications, I am happy to involve in discussions.

---

> > > > > ### Author Response · Authors · 2024-12-03
> > > > >
> > > > > Dear Reviewer 9P4o,
> > > > >
> > > > > Thank you for your feedback. We now fully understand your concern on the ablation results and would like to make the following clarifications.
> > > > >
> > > > > First, it is worth mentioning that, following the original main paper, the previous ablation results we provided in Table R1/R2 is conducted **only on one of the test setting**, which happens to be the one with the minimum performance improvement among all our test settings. This is clear when comparing the results in Table 1, Table 2, and Table 3 in the main paper. Therefore, as the reviewer pointed out, the ablation study on this test setting may not sufficiently reveal the advantage of our complementary masking. We sincerely thank the reviewer for this reminder.
> > > > >
> > > > > To demonstrate the full potential of our method in a more comprehensive way, we have made every effort to conduct additional ablation studies on other representative test settings we have used across different types of images. We believe that the following tables clearly demonstrate the improvement of our complementary masking over the random masking baseline.
> > > > >
> > > > >
> > > > > ---
> > > > >
> > > > > Table R5. Ablation study of complementary masked loss and random masked loss on natural datasets, i.e., SYNTHIA → Cityscapses. We additionally provide the most competitive baseline (MIC) from Table 1 in the main paper.
> > > > >
> > > > > |       | mIoU  |
> > > > > |-------|--------|
> > > > > | MIC  | 74.0 |
> > > > > | Ours w/ random masked loss | 75.6 |
> > > > > | Ours w/ complementary masked loss | 76.7 |
> > > > >
> > > > >
> > > > > ---
> > > > >
> > > > > Table R6. Ablation study of complementary masked loss and random masked loss on VNC III → Lucchi (Subset1). We additionally provide the most competitive baseline (CAFA) from Table 2 in the main paper. **Note that IoU is the leading metric, instead of mAP.**
> > > > >
> > > > > |       | IoU       | mAP       | F1        | MCC      |
> > > > > |-------|-----------|-----------|-----------|----------|
> > > > > | CAFA  | 71.8      | 91.1      | 83.4      | 82.8     |
> > > > > | Ours w/ random masked loss | 73.6 | 91.6 | 84.7 | 84.1 |
> > > > > | Ours w/ complementary masked loss | 75.0 | 92.4 | 85.6 | 85.1 |
> > > > > ---
> > > > >
> > > > > Table R7. Ablation study of complementary masked loss and random masked loss on 3D synapse detection, i.e., the WASPSYN Challenge dataset. We additionally provide the most competitive baseline (MIC) from Table 3 in the main paper.
> > > > >
> > > > > |       | $F1_{pre}$ | $F1_{pre}$ | F1-score  |
> > > > > |-------|-----------|-----------|-----------|
> > > > > | MIC | 0.7823 | 0.3599 | 0.5711 |
> > > > > | Ours w/ random masked loss | 0.7778 | 0.3792 | 0.5785 |
> > > > > | Ours w/ complementary masked loss | 0.7914 | 0.3912 | 0.5913 |
> > > > > ---
> > > > >
> > > > > These results comprehensively demonstrate the superiority of our method, and we hope your concerns on the ablation study can be adequately addressed.
> > > > >
> > > > > Last but not least, as the reviewer suggested, we will also highlight our **theoretical contributions**, which are an indispensable part of our work and are **entirely lacking in existing works**. We believe our theoretical work would provide insights for bridging Masked Image Modeling and Unsupervised Domain Adaptation.
> > > > >
> > > > > Once again, thanks for your time and dicsussion. We look forward to your final justification within the last day of dicussion.

---

### Official Review · Reviewer_E8tE · 2024-11-04

**Soundness:** 3
**Presentation:** 3
**Contribution:** 2
**Rating:** 3
**Confidence:** 4

**Summary:**

The paper proposes MaskTwins for unsupervised domain-adaptive image segmentation. The key method is dual-form complementary masking, where masked image modeling (MIM) is used to generate dual complementary views of images. This approach enables robust feature learning by enforcing prediction consistency across complementary masked images, allowing for adaptation to target domains without additional learnable parameters. MaskTwins demonstrates strong performance improvements over previous unsupervised domain adaptation (UDA) techniques, achieving state-of-the-art results on diverse datasets, including both natural and biological images.

**Strengths:**

- This paper provides a solid theoretical basis for the use of dual-form complementary masks.

- The method could be easily integrated into existing frameworks with minimal overhead.

**Weaknesses:**

- This paper is similar to [1*]. Therefore, it seems that the complementary masking techniques are not novel.

(1) Both papers propose complementary masking techniques to promote robustness.

(2) Both methods use consistency mechanisms to enforce feature learning.

[1*] Shin U, Lee K, Kweon I S, et al. Complementary random masking for rgb-thermal semantic segmentation[C]//2024 IEEE International Conference on Robotics and Automation (ICRA). IEEE, 2024: 11110-11117.

- The method relies on pseudo-labels generated by an exponential moving average (EMA) teacher model for unsupervised target domain training. This dependence on pseudo-label quality could introduce noise into training if the initial pseudo-labels are inaccurate.

- The ablation study of different components in the model should be provided.

**Questions:**

See section weakness

---

> ### Author Response · Authors · 2024-11-22
>
> Thank you for your valuable time and comments. The main concerns are addressed below.
>
> > **W1:** This paper is similar to [1]. Therefore, it seems that the complementary masking techniques are not novel.
>
> > (1) Both papers propose complementary masking techniques to promote robustness.
>
> Thank you for mentioning [1], which is an inspiring work and we will cite it in the revised paper. However, our work is inherently different from [1] in terms of the masking definition and the application scope. More importantly, we have provided theoretical analysis of our complementary masking strategy.
>
> 1. The masking definition is different. [1] performs complementary masking strategy on two modalities: RGB and thermal images. In contrast, we focus on a single modality to ensure that **both the masks and the generated masked images** are complementary. Therefore, the definition of complementary masking in our work is more flexible.
> 2. The application scope is different. [1] focuses on RGB-Thermal semantic segmentation, tackling the over-reliance on one modality in multimodal inputs. The masking technique in [1] is tightly coupled with RGB-thermal pairs and relies on these specific multimodal datasets. In contrast, our work addresses the unsupervised domain adaptation (UDA) problem, which involves adapting models across domains with unlabeled target-domain data. Our complementary masking strategy is applied to single-modality inputs, which makes it more generalizable and widely applicable across tasks and datasets.
> 3. We have strictly proved that complementary masking can effectively extract domain-invariant features. In our work, complementary masking is framed as a dual-form masking strategy grounded in sparse signal reconstruction theory. This theoretical foundation is a core contribution of our work, as it rigorously proves the advantages of complementary masking in preserving information, reducing variance, and improving feature consistency.
>
> >(2) Both methods use consistency mechanisms to enforce feature learning.
>
> The "consistency mechanism" is a generic concept and does not indicate significant similarity. Consistency mechanisms are widely used in semi-supervised learning and UDA tasks as a fundamental strategy to promote robust feature learning. While both works adopt consistency mechanisms, our work leverages dual-form complementary masking consistency as a novel way to enforce feature learning across complementary views, specifically tailored for unsupervised domain adaptation. This is a natural progression in UDA tasks and does not imply direct overlap with [1].
>
> In conclusion, while [1] provides a practical solution for RGB-Thermal segmentation, our work builds a rigorous theoretical framework for complementary masking and applies it to the broader, more challenging problem of domain adaptation. We appreciate the opportunity to clarify these distinctions in the revised paper.
>
> > **W2:** The dependence on pseudo-labels
>
>
> We agree that pseudo-labeling introduces inherent noise due to the distribution gap between the source and target domains. However, this is a well-established challenge in UDA tasks. Addressing this, we emphasize the following points:
> 1. Pseudo-label quality is not the core focus of our work. To avoid distracting from our main contributions—complementary masking and consistency learning—we adopt the EMA teacher model for pseudo-label generation, following common practices in UDA methods such as CAMix [2], MIC [3], and DAFormer [4]. This ensures fair and consistent comparison with existing state-of-the-art methods.
> 2. Our complementary masked loss reduces dependence on pseudo-labels. Unlike traditional UDA methods that rely heavily on pseudo-label quality for learning, our dual-form complementary masking strategy introduces an additional component of robustness. Specifically, the complementary masked loss encourages the model to make consistent predictions across masked views without the supervision of pseudo-label, which mitigates the impact of noise during training. This is evident in Table R1, where the complementary masked loss improves segmentation performance even in the challenging UDA scenario.
> 3. EMA pseudo-labeling is widely adopted and effective for UDA. While pseudo-labeling introduces noise, it has been shown to be an effective strategy when combined with consistency regularization. Our work builds on this foundation and demonstrates that complementary masking further enhances robustness, as reflected in our experimental results.

---

> ### Author Response · Authors · 2024-11-22
>
> > **W3:** Ablation study of the used components
>
> As suggested, we fully ablate the used components on MitoEM-H → MitoEM-R fully in the following tables.
>
> ---
>
> Table R1. Ablation study of each loss component on MitoEM-H → MitoEM-R. The mean and standard deviation are computed over 3 random seeds. $L_{sup}$ = supervised loss, $L_{cl}$ = consistency loss, $L_{cm}$ = complementary masked loss, $L_{rm}$ = randomly masked loss, with a mask ratio of 0.5.
>
> |       | mAP       | F1        | MCC       | IoU       |
> |-------|-----------|-----------|-----------|-----------|
> | $L_{sup}$  | 96.38(±0.18) | 88.95(±0.07) | 88.60(±0.07) | 80.11(±0.11) |
> | $L_{sup}+L_{cl}$ | 96.60(±0.35) | 89.27(±0.10) | 88.94(±0.12) | 80.64(±0.17) |
> | $L_{sup}+L_{cl}+L_{rm}$ | 96.84(±0.10) | 89.80(±0.03) | 89.45(±0.04) | 81.49(±0.04) |
> | $L_{sup}+L_{cl}+L_{cm}$ | 96.87(±0.06) | 90.03(±0.06) | 89.66(±0.04) | 81.88(±0.09) |
> ---
>
>
> As shown in Table R1, adding the consistency loss $L_{cl}$ improves the performance over the supervised loss. Further, we separately incorporate the randomly masked loss $L_{rm}$ and the complementary masked loss $L_{cm}$. The results indicate that both losses contribute to performance improvements, with our proposed complementary masking strategy being more effective than the random masking strategy. The results with $L_{rm}$ and $L_{cm}$ have been visually shown in Figure 4 in the main paper.
>
> ---
>
> Table R2. Ablation study of EMA and AdaIN on MitoEM-H → MitoEM-R. The mean and standard deviation are computed over 3 random seeds.
>
> |       | mAP       | F1        | MCC       | IoU       |
> |-------|-----------|-----------|-----------|-----------|
> | Ours w/o  EMA & AdaIN | 96.74(±0.08) | 89.61(±0.05) | 89.23(±0.04) | 81.18(±0.08) |
> | Ours w/o EMA  | 96.85(±0.15) | 89.74(±0.02) | 89.38(±0.04) | 81.40(±0.04) |
> | Ours w/o AdaIN | 96.89(±0.14) | 89.88(±0.05) | 89.52(±0.03) | 81.63(±0.08) |
> | Ours | 96.87(±0.06) | 90.03(±0.06) | 89.66(±0.04) | 81.88(±0.09) |
> ---
>
> The results show that both EMA and AdaIN contribute to performance improvements, with EMA having a more significant impact.
>
> **Reference**
>
> [1] Shin, Ukcheol, et al. "Complementary random masking for rgb-thermal semantic segmentation." 2024 IEEE International Conference on Robotics and Automation (ICRA). IEEE, 2024.
>
> [2] Zhou, Qianyu, et al. "Context-aware mixup for domain adaptive semantic segmentation." IEEE Transactions on Circuits and Systems for Video Technology 33.2 (2022): 804-817.
>
> [3] Hoyer, Lukas, et al. "MIC: Masked image consistency for context-enhanced domain adaptation." Proceedings of the IEEE/CVF conference on computer vision and pattern recognition. 2023.
>
> [4] Hoyer, Lukas, Dengxin Dai, and Luc Van Gool. "Daformer: Improving network architectures and training strategies for domain-adaptive semantic segmentation." Proceedings of the IEEE/CVF conference on computer vision and pattern recognition. 2022.

---

> > ### Comment · Reviewer_E8tE · 2024-11-26
> > **Thank you for the rebuttal**
> >
> > Thanks for their detailed rebuttal and clarifications. While I appreciate the efforts to address the raised concerns, I believe the proposed method remains an incremental improvement over existing approaches. Despite some differences highlighted in the response, the methodological contributions appear to be limited. Consequently, I maintain my original score as my perspective on the overall contribution of the work remains unchanged.

---

> > > ### Author Response · Authors · 2024-11-30
> > >
> > > Dear Reviewer E8tE,
> > >
> > > Thank you for your valuable time and comments. But we respectfully cannot agree with the justifications here, and would like to further clarify as below:
> > >
> > > We highlight our **theoretical contributions**, which are an indispensable part of our work and are **entirely lacking in existing works**. Specifically, we have provided a theoretical foundation for masked reconstruction from a perspective of sparse signal reconstruction problem and have rigorously analyzed the properties of complementary masking from three aspects: **information preservation, generalization bound, and feature consistency**.
> > >
> > > Since these properties are critical for extracting domain-invariant features, we conduct comprehensive experiments to validate the effectiveness of complementary masking on unsupervised domain adaptation (UDA) tasks, from natural to biological image segmentation. Experimental results demonstrate that the improvement brought about by our complementary masking is greater than that of other well-constructed components used, as shown in Table R2.
> > >
> > > It is worth mentioning that an excessive accumulation of methodological components (the way a lot of other works are done nowadays) could lead to a diversion from our core theoretical contributions and potentially dilute the validity of our experimental results. Therefore, we choose to maintain the simplicity of our approach and to strictly adhere to our theoretical assumptions, as **"simple yet effective" is more meaningful than "complicated and effective".**
> > >
> > > In summary, **our work is rigorous and comprehensive, spanning from theory to experimentation.** We believe our theoretical work would provide insights for bridging Masked Image Modeling and UDA.
> > >
> > > If there are any remaining issues, we are happy to provide more clarifications. Otherwise, we look forward a more positive feedback.

---

### Official Review · Reviewer_2eV4 · 2024-11-04

**Soundness:** 3
**Presentation:** 3
**Contribution:** 3
**Rating:** 8
**Confidence:** 3

**Summary:**

The paper introduces a new methodology based on masked image modeling for unsupervised domain adaptation. The paper shows theoretically that the complementary masking strategy outperforms random masking on information preservation, generalization and feature consistency. The proposed methodology is evaluated on natural image segmentation, mitochondria semantic segmentation and synapse detection.

**Strengths:**

The paper is overall very enjoyable to read. In particular, i find that:

* The paper is clearly written, well-situated in the literature and is easy to read and understand.
* The proposed methodology is original and constitutes and important bridge between SSL and UDA.
* The proposed methodology excels in its simplicity, in particular compared to adversarial UDA frameworks. In particular, i find it noteworthy that the proposed methodology is effective on 3D input (synapse detection), in which it is generally notoriously hard to stabilise UDA training.
* The evaluation is thorough and includes comparisons to a wide range of UDA baselines.
* The theoretical foundation is sufficient for the study’s objectives, but could be strengthened to enhance rigor.

**Weaknesses:**

The main short comings of the paper seems to be the limited scope of the theoretical analysis and the ablation study.

* First the theoretical analysis does not connect the proposed methodology to the vast literature on the theory for unsupervised domain adaptation, in particular [1], [2], [3]. Instead the theoretical results relate the methodology to a naive masking strategy, which might still provide benefits over other methodologies.

* Second, the ablation study does not study the effectiveness of individual parts of the methodology. I find that it is central to include a comparison to a simpler masking strategy, such as the random baseline from the theoretical analysis and to study the impact of each element in the loss. Instead the authors choose to only ablate details on the patch size, mask type and masking ratio.

References:

[1] Shai Ben-David, John Blitzer, Koby Crammer, and Fernando Pereira. Analysis of repre- sentations for domain adaptation. In NIPS, pages 137–144, 2006.

[2] Shai Ben-David, John Blitzer, Koby Crammer, Alex Kulesza, Fernando Pereira, and Jen- nifer Wortman Vaughan. A theory of learning from different domains. Machine Learning, 79(1-2):151–175, 2010.

[3] Zhang, Yuchen, et al. "Bridging theory and algorithm for domain adaptation." _International conference on machine learning_. PMLR, 2019.

**Questions:**

Ablation study:

* How does the framework perform if the naive masking strategy is used compared to complementary masking? (I.e. no complementary loss). And more generally, what is the performance gain of each element in the loss? (i.e. complementary vs. consistency vs. supervised loss.)
* What is the benefit of using an EMA teacher-student framework and what is the effect of not including AdaIN?

Evaluation:

*  How many folds/runs are the evaluation results based on?
*  How stable is the training? (i.e. the variance of the runs.)


Overall i find that the paper is important and I am inclined to reconsider my score if above points are addressed in the rebuttal.

---

> ### Author Response · Authors · 2024-11-22
>
> Thank you for your valuable time and comments. The main concerns are addressed below.
>
> >**W1:** Connection to UDA in the theoretical analysis
>
> As suggested, we will briefly discuss how our complementary masking strategy relates to the theory of domain adaptation in the revised paper to further enrich our theoretical background.
>
> The theoretical works [1, 2, 3] provide fundamental insights into UDA, especially concerning domain discrepancy and theoretical bounds. Specifically, they study margin bounds for classification tasks at the distribution level, while we focus on segmentation tasks and the theory of Masked Image Modeling and compressed sensing at the image level. We have analyzed the information preservation, generalization bounds and feature consistency to demonstrate the effectiveness of complementary masking. The paper [3] which is grounded in the theory of domain adaptation such as [1] and [2] also discusses generalization bounds based on empirical Rademacher complexity. We preliminary observe that there exists deeper connections between these works and ours. Hopefully, we will make further theoretical analysis in the next version.
>
> As for the "naive masking strategy", it serves as a control group to ensure a fair comparison with our complementary mask. We have included comparisons to the random masking strategy (widely adopted in mask-related works such as MAE) in our ablation study, demonstrating that our complementary masking provides significant benefits over random masking. We validate in the mask type section of our ablation study the significant advantages of complementary masking compared to random masking.
>
>
> > **W2 & Q1.1:** Comparison with random baseline and ablation study of loss components
>
> First, we have compared the complementary masking with the random baseline from the theoretical analysis by conducting an ablation study on mask types in the main paper, and we provide the results again below. Next, as suggested, we fully ablate different components of losses on MitoEM-H → MitoEM-R in the following table.
>
> ---
>
> Table R1. Ablation study of each loss component on MitoEM-H → MitoEM-R. The mean and standard deviation are computed over 3 random seeds. $L_{sup}$ denotes the supervised loss, $L_{cl}$ is the consistency loss, $L_{cm}$ is the complementary masked loss, and $L_{rm}$ is the randomly masked loss, with a mask ratio of 0.5.
>
> |       | mAP       | F1        | MCC       | IoU       |
> |-------|-----------|-----------|-----------|-----------|
> | $L_{sup}$  | 96.38(±0.18) | 88.95(±0.07) | 88.60(±0.07) | 80.11(±0.11) |
> | $L_{sup}+L_{cl}$ | 96.60(±0.35) | 89.27(±0.10) | 88.94(±0.12) | 80.64(±0.17) |
> | $L_{sup}+L_{cl}+L_{rm}$ | 96.84(±0.10) | 89.80(±0.03) | 89.45(±0.04) | 81.49(±0.04) |
> | $L_{sup}+L_{cl}+L_{cm}$ | 96.87(±0.06) | 90.03(±0.06) | 89.66(±0.04) | 81.88(±0.09) |
> ---
>
>
> As shown in Table R1, adding the consistency loss $L_{cl}$ improves the performance over the supervised loss. Further, we separately incorporate the randomly masked loss $L_{rm}$ and the complementary masked loss $L_{cm}$. The results indicate that both losses contribute to performance improvements, with our proposed complementary masking strategy being more effective than the random masking strategy. The results with $L_{rm}$ and $L_{cm}$ have been visually shown in Figure 4 in the main paper.
>
>
> > **Q1.2:** The benefit of EMA teacher-student framework and the effect of not including AdaIN
>
>
> The EMA teacher realizes a temporal ensemble of previous student models, which increases the robustness and temporal stability of pseudo-labels. It is a common strategy used in semi-supervised learning and UDA. In our work, we adopt the EMA teacher to keep consistent with previous methods, such as CAMix [4], MIC [5], DAFormer [6], etc.
>
> We further ablate the effect of the EMA teacher and AdaIN in the following table. The results show that both EMA and AdaIN contribute to performance improvements, with EMA having a more significant impact.
>
>
> ---
>
> Table R2. Ablation study of EMA and AdaIN on MitoEM-H → MitoEM-R. The mean and standard deviation are computed over 3 random seeds.
>
> |       | mAP       | F1        | MCC       | IoU       |
> |-------|-----------|-----------|-----------|-----------|
> | Ours w/o  EMA & AdaIN | 96.74(±0.08) | 89.61(±0.05) | 89.23(±0.04) | 81.18(±0.08) |
> | Ours w/o EMA  | 96.85(±0.15) | 89.74(±0.02) | 89.38(±0.04) | 81.40(±0.04) |
> | Ours w/o AdaIN | 96.89(±0.14) | 89.88(±0.05) | 89.52(±0.03) | 81.63(±0.08) |
> | Ours | 96.87(±0.06) | 90.03(±0.06) | 89.66(±0.04) | 81.88(±0.09) |
> ---

---

> ### Author Response · Authors · 2024-11-22
>
> > **Q2.1:** The folds/runs of the evaluation results
>
> Our results are averaged over 3 random seeds, following [5, 6]. We do not use cross validation since the target domain (test set) is fixed in the setting of UDA tasks.
>
> > **Q2.2:** The stability of training
>
> We provide the mean and standard deviation of the experiments as below. For natural image segmentation and mitochondria segmentation, we are not able to include comparison results with other methods because their original papers did not report precise standard deviations, and some methods have not been open-sourced. For synapse detection, we provide the results of other methods since these are reproduced by us. The results indicate that our method demonstrates high stability across different runs, with low variance in performance metrics. This demonstrates the robustness and reliability of our training process.
>
> ---
>
> Table R3. Natural Image Segmentation results on SYNTHIA → Cityscapses. For more details, see Table 1.
>
> |  | Road | SW  | Build | TL  | TS  | Veg. | Sky | PR  | Rider | Car | Bus | Motor | Bike | mIoU |
> |--------|------|-----|-------|-----|-----|------|-----|-----|-------|-----|-----|-------|------|------|
> | Ours   | 96.0(±0.2) | 70.1(±1.2) | 89.5(±0.1) | 66.8(±0.4) | 62.1(±1.0) | 89.1(±0.1) | 94.3(±0.2) | 81.5(±0.2) | 59.7(±1.2) | 90.5(±0.1) | 66.6(±0.8) | 67.7(±0.3) | 63.6(±1.0) | 76.7(±0.2) |
> ---
>
> Table R4. Mitochondria Segmentation results. For more details, see Table 2.
>
> |       | mAP       | F1        | MCC       | IoU       |
> |-------|-----------|-----------|-----------|-----------|
> | VNC III → Lucchi (Subset1)  | 92.4(±0.1) | 85.6(±0.1) | 85.1(±0.1) | 75.0(±0.3) |
> | VNC III → Lucchi (Subset2) | 95.2(±0.1) | 87.9(±0.3) | 87.4(±0.2) | 78.6(±0.2) |
> | MitoEM-R → MitoEM-H | 94.0(±0.0) | 87.9(±0.2) | 87.4(±0.1) | 78.4(±0.3) |
> | MitoEM-H → MitoEM-R | 96.9(±0.1) | 90.0(±0.1) | 89.7(±0.0) | 81.9(±0.1) |
> ---
>
> Table R5. Synapse Detection results.
>
> |       | $F1_{pre}$ | $F1_{pre}$ | F1-score  |
> |-------|-----------|-----------|-----------|
> | SSNS-Net  | 0.7201(±0.0059) | 0.3072(±0.0050) | 0.5137(±0.0054) |
> | AdaSyn | 0.7846(±0.0065) | 0.3136(±0.0108) | 0.5491(±0.0032) |
> | MIC | 0.7823(±0.0078) | 0.3599(±0.0069) | 0.5711(±0.0026) |
> | Ours | 0.7914(±0.0080) | 0.3912(±0.0057) | 0.5913(±0.0021) |
> ---
>
> **Reference**
>
> [1] Shai Ben-David, John Blitzer, Koby Crammer, and Fernando Pereira. Analysis of representations for domain adaptation. In NIPS, pages 137–144, 2006.
>
> [2] Shai Ben-David, John Blitzer, Koby Crammer, Alex Kulesza, Fernando Pereira, and Jennifer Wortman Vaughan. A theory of learning from different domains. Machine Learning, 79(1-2):151–175, 2010.
>
> [3] Zhang, Yuchen, et al. "Bridging theory and algorithm for domain adaptation." International conference on machine learning. PMLR, 2019.
>
> [4] Zhou, Qianyu, et al. "Context-aware mixup for domain adaptive semantic segmentation." IEEE Transactions on Circuits and Systems for Video Technology 33.2 (2022): 804-817.
>
> [5] Hoyer, Lukas, et al. "MIC: Masked image consistency for context-enhanced domain adaptation." Proceedings of the IEEE/CVF conference on computer vision and pattern recognition. 2023.
>
> [6] Hoyer, Lukas, Dengxin Dai, and Luc Van Gool. "Daformer: Improving network architectures and training strategies for domain-adaptive semantic segmentation." Proceedings of the IEEE/CVF conference on computer vision and pattern recognition. 2022.

---

> > ### Comment · Reviewer_2eV4 · 2024-11-26
> >
> > I would like to thank you for the detailed answer. In light of your clarifications and further results, I have decided to change my score to 8.

---

> > > ### Author Response · Authors · 2024-11-27
> > >
> > > Thank you for your thoughtful feedback and for updating your score! Your review really helped us greatly in improving our paper. And we sincerely appreciate your time and effort in reviewing our paper and reading our comments.

---

### Author Response · Authors · 2024-11-26

We sincerely thank the reviewers for their valuable comments and suggestions, and we hope our responses adequately address your concerns. The revised version of the manuscript has been uploaded. Furthermore, we are happy to provide further details on any aspects of our responses that may require additional clarification or elaboration.

Once again, we appreciate the reviewers’ time and insightful feedback and look forward to receiving further input.

---

### Author Response · Authors · 2024-12-04

We sincerely thank all the reviewers once again for their valuable time and feedback.

We have carefully incorporated each reviewer's constructive comments, making corresponding revisions and clarifying possible misunderstandings. We are encouraged to see that Reviewer 2eV4 have raised the score during discussion. Although not all reviewers have responded to our latest replies, we believe the major concerns should be adequately addressed.

We hope our efforts will be taken into consideration in your final justification.

---

### Meta-Review · Area_Chair_CWPs · 2024-12-22

**Metareview:**

The paper presents MaskTwins, leveraging dual-form complementary masking for domain-adaptive image segmentation, supported by a theoretical framework connecting masked image modeling to sparse signal reconstruction. While the theoretical contributions are appreciated, the reviewers consistently raised concerns about the lack of empirical validation to substantiate these contributions beyond standard benchmark evaluations. The limited scope of ablation studies and marginal performance improvements over random masking strategies also raised questions regarding the claims. Although the authors provided detailed responses and additional experiments during the rebuttal period, they were not able to fully address these issues, leaving questions about the practical impact of the theoretical insights. Therefore, further revisions are recommended for this work.

**Additional Comments On Reviewer Discussion:**

The reviewers expressed concerns about the lack of empirical validation for the theoretical contributions, the incremental novelty of the proposed complementary masking method, and the limited scope of the ablation studies. In response, the authors conducted additional ablation experiments, compared their method to random masking strategies, and presented analyses showcasing performance improvements across multiple datasets. They also clarified the theoretical foundation of their approach, highlighting its connection to sparse signal reconstruction and its purported advantages over existing techniques. However, the results demonstrated only marginal gains in performance, and the reviewers noted a disconnect between the theoretical contributions and the methodology, which they deemed incremental. While the authors attempted to bridge the theory with unsupervised domain adaptation (UDA), they did so without providing sufficient supporting analyses. This led to a score increase from 5 to 8 by reviewer 2eV4, but no changes in assessment by the other reviewers.

---

### Decision · Program_Chairs · 2025-01-22

Reject